# CBP/p300: Critical Co-Activators for Nuclear Steroid Hormone Receptors and Emerging Therapeutic Targets in Prostate and Breast Cancers

**DOI:** 10.3390/cancers13122872

**Published:** 2021-06-08

**Authors:** Aaron R. Waddell, Haojie Huang, Daiqing Liao

**Affiliations:** 1UF Health Cancer Center, Department of Anatomy and Cell Biology, University Florida College of Medicine, 2033 Mowry Road, Gainesville, FL 32610, USA; aawaddell@ufl.edu; 2Departments of Biochemistry and Molecular Biology and Urology, Mayo Clinic College of Medicine and Science, 200 First St. SW, Rochester, MN 55905, USA; Huang.Haojie@mayo.edu

**Keywords:** CBP/p300, acetyltransferase, bromodomain, histone acetylation, transcription co-activator, cancer epigenetics, breast cancer, prostate cancer

## Abstract

**Simple Summary:**

The CREB-binding protein (CBP) and p300 are paralogous lysine acetyltransferases that serve as critical co-activators for transcription factors involved in diverse signaling pathways in cancer. Work in the last two decades has firmly established CBP and p300 as important regulators of nuclear hormone signaling mediated by nuclear receptors, such as the androgen receptor (AR) and estrogen receptor (ER). The AR and ER promote tumor growth in hormone-dependent prostate and breast cancer, respectively. Inhibitors of androgen and estrogen signaling are the standard-of-care therapeutics for treating these cancers. However, resistance to current therapies remains a significant clinical problem. Inhibition of CBP and p300 as a means to block the transactivation activity of the AR and ER is an emerging therapeutic strategy for prostate and breast cancers. This review describes how CBP and p300 regulate androgen and estrogen signaling and discusses therapeutic potential of newly discovered potent CBP/p300 inhibitors for treating prostate and breast cancer.

**Abstract:**

The CREB-binding protein (CBP) and p300 are two paralogous lysine acetyltransferases (KATs) that were discovered in the 1980s–1990s. Since their discovery, CBP/p300 have emerged as important regulatory proteins due to their ability to acetylate histone and non-histone proteins to modulate transcription. Work in the last 20 years has firmly established CBP/p300 as critical regulators for nuclear hormone signaling pathways, which drive tumor growth in several cancer types. Indeed, CBP/p300 are critical co-activators for the androgen receptor (AR) and estrogen receptor (ER) signaling in prostate and breast cancer, respectively. The AR and ER are stimulated by sex hormones and function as transcription factors to regulate genes involved in cell cycle progression, metabolism, and other cellular functions that contribute to oncogenesis. Recent structural studies of the AR/p300 and ER/p300 complexes have provided critical insights into the mechanism by which p300 interacts with and activates AR- and ER-mediated transcription. Breast and prostate cancer rank the first and forth respectively in cancer diagnoses worldwide and effective treatments are urgently needed. Recent efforts have identified specific and potent CBP/p300 inhibitors that target the acetyltransferase activity and the acetytllysine-binding bromodomain (BD) of CBP/p300. These compounds inhibit AR signaling and tumor growth in prostate cancer. CBP/p300 inhibitors may also be applicable for treating breast and other hormone-dependent cancers. Here we provide an in-depth account of the critical roles of CBP/p300 in regulating the AR and ER signaling pathways and discuss the potential of CBP/p300 inhibitors for treating prostate and breast cancer.

## 1. Introduction

The CREB-binding protein (CBP, also known as KAT3A, encoded by the CREBBP gene) and p300 (also known as KAT3B, encoded by the EP300 gene) are two paralogous lysine acetyltransferases (KATs) that were discovered in the 1980s–1990s [1,2,3,4]. CBP and p300 share a highly conserved modular structure. Their acetyltransferase domain (HAT or KAT; HAT is referred to hereafter), acetyllysine-binding bromodomain (BD), and several structured modules (e.g., KIX, TAZ1, TAZ2, and iBID) are virtually identical (Figure 1A). For example, the HAT domains of CBP and p300 are 89% identical and 93% similar. Thus, CBP and p300 were originally believed to have exchangeable functions and are often referred to together as CBP/p300 [5,6]. In this context, it is important to note that the amino acid sequences of these two proteins are highly divergent in regions between their conserved domains, suggesting CBP/p300 may be differentially regulated and have non-redundant functional outputs. Indeed, context-dependent roles for p300 and CBP have been observed in embryonic development and tumorigenesis [7,8,9,10,11].

CBP/p300 are important regulators of gene expression and are recruited to chromatin through interacting with many DNA-binding transcription factors [12,13,14]. CBP/p300 were initially thought to act as a bridge between enhancers and the RNA polymerase machinery to regulate transcription [13]. However, the pivotal discovery that CBP/p300 has HAT activity shed new light on their role in gene regulation. Historically, histone acetylation and its role in gene regulation were discovered over 57 years ago [15,16]. The enzymes that catalyze histone acetylation, including CBP/p300, were first reported in 1996 [17,18,19,20,21]. It is now clear that CBP/p300 catalyzes lysine acetylation on a broad range of substrates, including both histone and many non-histone proteins, to regulate numerous signaling pathways involved in cell growth, development, and tumorigenesis [12,22]. CBP/p300 and the histone acetylation sites they modify are now recognized as functionally important for active promoters and enhancers [23,24]. However, the functional impact of CBP/p300 acetylation of non-histone substrates is only beginning to be investigated. The determination of the structures of the HAT and the core domain encompassing the BD, RING-PHD and the HAT regions of CBP/p300 (Figure 1A) provides the structural basis for our understanding of the mechanisms underlying CBP/p300-mediated acetylation reaction and for the designs of chemical inhibitors of HAT activity [5,25,26,27,28].

Early work on p300 identified this protein as a potential tumor suppressor because viral oncoproteins, such as the adenovirus E1A protein, inhibit p300 activity to promote cell transformation [29,30]. In addition, loss-of-function genetic mutations of the EP300 gene (encoding p300) and CREBBP (encoding CBP) in cancer suggest that CBP/p300 could serve as a tumor suppressor [31,32]. Chromosome translocations of both EP300 and CREBBP in cancer have also been observed [33,34]. Recent studies, however, reveal that CBP/p300 can promote tumorigenesis. p300 has been shown to acetylate the fusion oncoprotein AML1-ETO to promote leukemogenesis [35]. Inhibition of p300 activity exhibits synthetic lethal effects on tumors deficient of CREBBP due to MYC downregulation [36], highlighting a critical role of having at least one functional paralog of these two proteins for cancer cell survival. Tumor-specific glucose fermentation enhances the recruitment of CBP/p300 to CCND1 gene to promote tumor cell proliferation [37]. Notably, a recent global chromatin profiling study showed that truncating mutations of CREBBP or EP300 within their TAZ2 domain correlate with increased levels of histone H3 acetylated at lysines 27 and 18 (H3K27ac and H3K18ac) in a number of cancer cell lines derived from different cancer types, suggesting that these mutations activate the enzymatic activity of CBP/p300 [38]. Proteogenomic profiling of clinical breast tumor samples provides evidence of CBP activation and elevated acetylation of many substrates in breast cancer (BC) [39]. These works shed important light on the oncogenic roles of CBP/p300, but much work remains to be done to elucidate their contribution to tumor progression in different cancer types.

A significant amount of research now shows that CBP/p300 is critical for pro-growth nuclear steroid hormone receptor signaling in prostate cancer (PCa) and estrogen receptor-positive (ER+) BC. Androgen receptor-positive (AR+) PCa and ER+ BC are dependent on oncogenic hormone signaling mediated by the AR and the ER, respectively [40,41]. In hormone-dependent PCa and BC, the AR and ER function as transcription factors to regulate genes involved in cell cycle progression, metabolism, and other cellular functions to promote tumor growth [42,43,44,45,46,47,48,49]. CBP/p300 serve as co-activators for both nuclear receptors to activate their target genes through interacting with the AR or ER on chromatin. Table 1 summarizes key features of the interactions of CBP/p300 with the AR and ER.

PCa and ER+ BC represent large global health burdens and hormone therapies targeting the AR and ER are standard treatments, but clinical resistance to these therapies is common [40,41,56,57]. In PCa, androgen deprivation therapy (ADT) blocks AR activation in patients; however, tumors eventually relapse and progress to metastatic disease [41,57]. Newer drugs, such as AR antagonist enzalutamide, only extend the survival of PCa patients by several months and nearly all patients develop resistance [57]. Similarly, 20–30% of ER+ BC cases are resistant to ER antagonist tamoxifen, the standard-of-care treatment for these patients [56]. Resistance to hormone therapies can occur through several mechanisms, such as activation of the AR and ER via alternative signaling pathways and the emergence of drug-resistant AR and ER mutants [41,49,57,58,59,60,61,62]. Therefore, new strategies are needed for inhibiting oncogenic hormone signaling and overcoming treatment resistance in these cancer types.

The realization that CBP/p300 is crucial for AR and ER signaling has led to significant interest in targeting CBP/p300 as an alternative therapeutic strategy to treat PCa and BC. Several specific and potent inhibitors of CBP/p300 function have been discovered. These inhibitors effectively block AR and ER activity and tumor growth. This review aims to provide an in-depth account of the roles of CBP/p300 in regulating hormone signaling and the application of recently developed CBP/p300 inhibitors in hormone-dependent PCa and their potential application in BC. Additionally, common and distinct functions for CBP and p300 in gene regulation and tumorigenesis will be discussed [10,11,53,63].

## 2. CBP/p300 Are Emerging Therapeutic Targets with Pharmacologically Tractable HAT Domain and Bromodomain

### 2.1. CBP/p300 Acetylate Diverse Proteins and Their HAT Domain Represents a Druggable Target

The modular structure of CBP/p300 consists of eight distinct domains: the nuclear receptor interaction (NRID), C/H1, KIX, bromo, C/H2, the enzymatic lysine acetyltransferase (HAT), C/H3 (which also contains the ZZ and TAZ2 domain), and the nuclear co-activator-binding (NCBD) domains (Figure 1A) [12,27,64]. Overall, CBP/p300 are 61% similar in sequence and the bromodomain-HAT-CH3 (BHC) region is 86% similar [12].

CBP/p300 catalyze the transfer of an acetyl group from acetyl-Coenzyme A (acetyl-CoA) to a lysine residue on substrates [12]. CBP/p300 catalyze acetylation on a broad range of substrates, from histones to many non-histone proteins [12,22]. To date, CBP/p300 is known to acetylate 20 distinct sites on the N-terminus of the four core histones (H2A, H2B, H3, and H4) (Figure 1B) [22]. Several of these acetylation sites, such as Histone 3 Lysine 18 and 27 (H3K18ac and H3K27ac, respectively), are enriched at transcriptionally active genomic regions, including promoters and enhancers [22,23,24,65,66,67,68,69,70]. As such, these acetylation markers are often used in ChIP-seq to map active regulatory regions [23,65,70]. Importantly, H3K27ac is now recognized as being functionally important for gene expression driven by enhancers and promoters and is no longer thought of as simply a marker for active regions [23,24]. These newer studies provide a deeper mechanistic understanding of how CBP/p300-catalyzed histone acetylation directly regulates gene expression.

Acetylation occurs on thousands of cellular proteins involved in diverse functions, such as the DNA damage response, transcription, cell signaling, and metabolism [71]. CBP/p300 also acetylates non-histone proteins to regulate their activity [72,73]. Shortly after p53 was identified as a substrate of p300 [74,75], the AR [76], ERα [77], NF-kB [78], and other transcription factors [13] were also reported to be modified by CBP/p300. Functionally, p300-mediated acetylation of p53 enhances p53’s DNA-binding and the activation of p53 target genes [74,79]. Acetylation of the AR by CBP/p300 is important for androgen-mediated AR activation [76]. Likewise, acetylation of ERα by CBP/p300 enhances ERα’s DNA-binding and transactivation function [80]. Acetylation of the AR and ER by CBP/p300 will be discussed further later in the review.

The important functional impact of CBP/p300 acetylation on diverse substrates has sparked investigation into the CBP/p300 acetylome. A recent mass-spectrometry study showed that CBP/p300 acetylates over 200 non-histone transcription factors and transcriptional co-activators [22]. Many of these substrates are involved in Wnt, notch, hippo, TGF-beta signaling, and nuclear hormone receptor signaling [22]. Additionally, CBP/p300 is known to interact with over 400 hundred protein partners through their folded structural domains (i.e., C/H1, KIX, bromodomain, C/H2, C/H3, and NCBD, Figure 1A) [12]. Therefore, CBP/p300 is capable of regulating diverse signaling pathways through their extensive interactome and catalysis of protein acetylation.

There have been continuous efforts to discover small molecule inhibitors of CBP/p300 catalytic activity, which have led to a number of chemical probes and drug candidates. These inhibitors include Lys-CoA, curcumin, anacardic acid, C646, A-485, and derivative compounds [12,28,81,82] (Table 2). Early inhibitor design was guided by CBP/p300’s catalytic mechanism. The CBP/p300 HAT domain has two substrates: acetyl-CoA and the lysine-containing target protein. This inspired early design of HAT inhibitors that structurally mimicked both substrates. Such inhibitors may be capable of tighter binding than either substrate alone. Lys-CoA is an example of a bi-substrate analog, which is a fusion of acetyl-lysine and coenzyme A moieties. The Lys-CoA inhibitor binds tightly to the CBP/p300 HAT domain and has proven valuable to study CBP/p300 structure/function [12]. The crystal structure of the p300 HAT domain in complex with Lys-CoA has been used for virtual ligand screens that dock compounds to the Lys-CoA binding pocket in p300 [28,83,84]. For example, C646 and A-485 were discovered using this approach.

A-485 is a highly potent and selective CBP/p300 HAT inhibitor [28]. A-485 is at least 1000-fold more potent than C646 in inhibiting CBP/p300 catalytic activity and does not inhibit other HATs up to 10 µM [28]. A-485 inhibits p300 and CBP catalytic activity with IC_50_ values of 9.8 and 2.6 nM, respectively [28]. A-485 competes with acetyl-CoA for access to the acetyl-CoA binding site in p300, making A-485 an acetyl-CoA competitive catalytic inhibitor of CBP/p300 [28]. A mass-spectrometry study showed that A-485 downregulates acetylation of 16 out of 20 histone sites regulated by CBP/p300, including H3K18ac and H3K27ac (Figure 1B) [22]. A-485 also strongly downregulates acetylation of non-histone substrates, including numerous signaling effectors in the notch, Wnt, and nuclear hormone receptor pathways [22]. Therefore, CBP/p300 HAT inhibition affects lysine post-translational modifications of proteins involved in diverse oncogenic signaling pathways.

### 2.2. The CBP/p300 Bromodomain Binds Acetylated Lysine Residues, Regulates HAT Activity, and Is a Target for Pharmacological CBP/p300 Inhibition

Bromodomains (BDs) are evolutionally conserved structural modules (approximately 110 amino acids) that specifically bind acetylated lysine residues on histones and non-histone proteins [94,95]. Structurally, BDs assume a bundle of four α-helices [95]. The BD is one of several epigenetic “reader” modules in epigenetic regulators that help to decode the “histone code” [94,96,97]. BDs are found in 46 different proteins in humans, including CBP/p300, and are under intense investigation for the development of small molecule inhibitors to block their function [94,98]. The first BD inhibitor (JQ1) was discovered in 2010 as a potent and specific ligand for members of the bromodomain and extra-terminal (BET) family [99].

The CBP/p300 BD facilitates protein–protein interactions that regulate transcription and other biological functions, such as DNA repair [100,101]. For example, acetylated E2F1 recruits CBP/p300 to DNA double-stranded breaks via a direct interaction with the CBP/p300 BD [100]. CBP/p300 induces H3 acetylation at sites of DNA damage and remodels chromatin structure to facilitate DNA repair [100]. Importantly, I-CBP112, a CBP/p300 BD inhibitor, appears to block CBP/p300 recruitment to sites of DNA damage [100]. Of note, the CBP/p300 BD appears to contribute to the formation of protein aggregates through acetyl-lysine binding, which could be mitigated by ligands specific for the CBP/p300 BD [102,103].

A number of CBP/p300 BD inhibitors have been discovered, including SGC-CBP30, I-CBP112, GNE-049, GNE-781, NEO2734, and Y08197 [52,91,104,105,106,107,108,109,110,111,112] (Table 2). To date, GNE-049 and GNE-781 are among the most potent and selective of the CBP/p300 BD inhibitors. Interestingly, GNE-049 reduces H3K27ac in cells (Figure 1B) [23], uncovering a role for the BD in regulating CBP/p300’s HAT activity [22,23,113,114,115]. GNE-049 impairs CBP/p300 HAT activity in test tube HAT reactions using CBP/p300 protein fragments and recombinant polynucleosomes [23]. Similar findings were shown in vitro with SGC-CBP30 [23]. This suggests that the BD has a direct role in regulating CBP/p300 HAT activity. Interestingly, the CBP BD was shown to interact with the acetylated autoregulatory loop within its HAT domain [27]. This intrinsically disordered autoregulatory loop is enriched with lysine residues that are autoacetylated by the HAT domain. Thompson et al. reported that autoacetylation of this loop activated p300’s HAT activity [116], while Park et al. showed that acetylation of K1596 impaired CBP’s HAT activity [27]. The deletion of the CBP BD and a peptide with acetylated K1596 inhibited H3K27 acetylation [27]. These observations suggest a possible scenario in which the acetylated autoregulatory loop binds to the BD of CBP/p300 intramolecularly to enable binding of their substrates for acetylation. BD inhibitors would block this intramolecular interaction, thus inhibiting acetylation of certain substrates. In another study, it was shown that transcription factor dimerization brings two molecules of p300 into proximity that enables autoacetylation of the autoregulatory loop in trans, thereby activating the HAT activity of p300 [117]. It is possible that BD inhibitors could also block the interaction of the autoregulatory loop with the CBP/p300 core domain in trans, which might impair their catalytic activity. Additional work will clearly be required to clarify how BD inhibitors impact the HAT activity of CBP/p300.

While CBP/p300 BD inhibitors downregulate HAT activity, they appear to affect fewer acetylation sites than HAT catalytic inhibitor A-485 or genetic knockdown of CBP/p300 (Figure 1B) [22,23]. For instance, I-CBP112 only reduces acetylation on 2 of 20 histone sites acetylated by CBP/p300 [22]. GNE-049 seems to have a broader inhibitory effect on HAT activity when tested in vitro, but GNE-049 only downregulates H3K27ac in cell culture (Figure 1B) [23]. GNE-049 also has minimal effect on the acetylation of non-histone proteins [23]. Therefore, CBP/p300 BD inhibitors appear to interfere with acetylation of specific acetylation sites in cells, rather than acting as general HAT inhibitors.

On a genomic scale, Raisner et al. proposed that the CBP/p300 BD is critical for H3K27ac deposition at enhancers because GNE-049 has a significantly greater inhibitory effect on H3K27ac at enhancers in comparison to promoters [23]. These findings led Raisner et al. to suggest that the CBP/p300 BD has an “enhancer-biased role” in histone acetylation [23]. Their work implies that the CBP/p300 BD may regulate HAT activity depending on genomic location. Further studies are required to fully elucidate the mechanism by which CBP/p300’s BD modulates genome-wide chromatin modifications.

Notably, some studies have reported that CBP/p300 BD inhibitors do not strongly displace endogenous CBP/p300 from chromatin, despite the BD being able to bind acetylated chromatin [23,106]. This suggests that binding of BDs to acetylated histones is not critical on a global scale for chromatin recruitment or retention of BD-containing proteins, possibly because other protein–protein interactions may be sufficient to tether BD proteins to chromatin [94,106]. However, other studies report that CBP/p300 BD inhibitors prevent CBP/p300 recruitment to specific genomic regions and can impact genome-wide CBP/p300 binding [100,118,119,120]. Therefore, the mechanism of action of CBP/p300 BD inhibitors may be attenuation of HAT activity, displacement of CBP/p300 from chromatin, or a combination of both mechanisms. Additional investigation is needed to determine why some studies report CBP/p300 genomic displacement with BD inhibitors, while others do not.

In summary, the ability of both CBP/p300 BD and HAT inhibitors to block the expression of oncogenes highlights the potential of these inhibitors as promising cancer therapeutics (Figure 1C). Notably, a recent study shows that a heterobifunctional molecule (dCBP-1) consisting of the CBP/p300 BD ligand GNE-781 [91] and a ligand of cereblon (CRBN) E3 ubiquitin ligase triggers the proteasomal degradation of CBP and p300 at a low nanomolar concentration [93]. dCBP-1 is more potent to abolish the enhancer function of CBP/p300 and to kill multiple myeloma cells than A-485 and GNE-781 [93]. Aside from showing the advantages of targeted protein degradation using dCBP-1 and numerous other proteolysis targeting chimeras (PROTACs) [121,122], this study further highlights the considerable therapeutic potential of pharmacological inhibition of CBP/p300 in oncology.

## 3. The Role of CBP/p300 in AR Signaling in PCa

### 3.1. PCa Is the Most Common Noncutaneous Cancer in Men and Is Predominantly Driven by the AR

Prostate cancer (PCa) represents an enormous global health burden, with an estimated 1.6 million cases and 366,000 deaths annually worldwide in men (Figure 2A) [41]. The AR plays a dominant role in PCa initiation and progression. The AR is a 110 kDa nuclear steroid hormone receptor that binds the male sex hormone testosterone and the more potent androgen 5α-dihydrotestosterone (DHT) [123,124]. The AR has three main domains: a long N-terminal unstructured activation domain (NTD), a short DNA-binding domain (DBD), and the C-terminal ligand-binding domain (LBD) [123]. In the absence of ligand, the AR is bound to heat shock protein (HSP) 90 and other chaperones in the cytoplasm (Figure 3A) [123,124]. Ligand-bound AR dimerizes, translocates to the nucleus, and binds androgen response elements (AREs) in the genome to regulate expression of its target genes (Figure 3A) [123,124]. Other transcription factors, such as the pioneering factor FOXA1, can enhance AR recruitment to ARE sites and impact AR complex formation (Figure 3A) [125,126].

The AR drives PCa growth through regulating the expression of genes involved in cell cycle progression, metabolism, and survival [44,46,48,49,127,128]. Of note, the AR also regulates cell cycle progression through mechanisms independent of transcription, such as enhancing translation of cyclin D mRNA and regulation of c-Myc protein levels [43,129]. Androgen deprivation therapies (ADTs) and new generation therapies targeting AR signaling (e.g., enzalutamide and abiraterone acetate [130,131]) are now the standard-of-care therapies for castration-resistant PCa (CRPC). However, both intrinsic and acquired resistance to these new drugs is a major clinical challenge (Figure 2B).

Mechanisms of treatment resistance in CRPC include AR gene amplification [132], mutations [133], expression of AR variants (AR-Vs) [134,135,136,137], AR bypass through glucocorticoid receptor (GR) activation [138], lineage plasticity [139,140,141,142,143], and co-regulator overexpression [144,145,146]. Interestingly, a recent study has shown that AR is also expressed in the majority of cases of treatment-resistant small-cell neuroendocrine prostate cancer (t-SCNC) [147]. However, it appears that AR target genes are epigenetically silenced and t-SCNC is indifferent to AR signaling blockade [148].

Various alternative strategies for AR inhibition, such as pharmacologic induction of AR protein degradation and targeting the AR N-terminal transactivation domain, have the potential to overcome drug resistance to current hormonal therapies [149,150]. Drug candidates based on these strategies have been tested clinically [151,152], but associated toxicities present a significant barrier to clinical application [152,153,154]. Therefore, resistance to AR-targeted therapies remains a persistent clinical problem [154] and innovative strategies are needed for suppressing AR function in late-stage recurrent CRPC. Continued efforts in understanding the biological and pathobiological function of the AR will be critical to developing innovative therapies to improve treatment outcomes for patients with CRPC.

### 3.2. p300 Is a Component of the AR Activation Complex

As a DNA-binding transcription factor, the AR has nearly 200 known co-regulators that can activate or repress AR target genes [124]. Targeting the co-regulators of the AR with small molecule inhibitors is an emerging strategy for ablating AR activity [28,52]. CBP/p300 are key co-activators for AR-mediated transcription and represent promising therapeutic targets to inhibit AR activity in PCa [50,155,156,157,158]. Early studies suggested that the AR recruits CBP/p300 to its target genes through an indirect interaction mediated by the Steroid Receptor Co-activator proteins (SRC-1, SRC-2, and SRC-3) [155,159,160]. Mechanistically it was thought that the LBD of the AR interacts with the SRC proteins, which then recruit CBP/p300 to the AR activation complex [155,159,160]. The SRC proteins interact with the LBD of nuclear receptors, including the AR, via an LXXLL motif found in the SRC receptor interaction domain (RID) (Figure 4) [159,160,161]. SRC-1 also interacts with a region known as the AF-1 in the NTD of the AR, which appears functionally more important than the SRC-1/AR-LBD interaction for AR-mediated transcriptional activation (Figure 4) [159].

CBP/p300 is recruited to nuclear receptor complexes through an interaction between SRC’s CBP/p300 interaction domain (CID) and CBP/p300’s SRC interaction domain (SID) (Figure 4) [51,155,162,163,164]. Notably, the LXXLL motif is also found in CBP/p300 for direct nuclear receptor binding [161]. However, direct interaction between CBP/p300 and nuclear receptors was shown to be weak [162]. The interaction between the SRC proteins and CBP/p300 is essential for the activity of the AR activation complex and contributes to histone acetylation at AR target genes, such as the KLK3 gene (encoding prostate-specific antigen or PSA) [155,159]. Of note, the co-activator-associated arginine methyltransferase 1 (CARM1) is also recruited to AR target genes upon DHT stimulation along with CBP/p300, which appears to methylate specifically H3R17 and R26 to activate gene expression [165].

This early model of AR/SRC/p300 interaction was further clarified in a recent cryoelectron microscopy (cryo-EM) structural study by Yu et al. [50]. Their work demonstrates that an androgen-activated AR homodimer binds the enhancer of the KLK3 gene, and each AR monomer directly interacts with p300 via their NTD (mainly AF-1) and small portions of the LBD (Figure 4) [50]. SRC-3 did not bridge the interaction between p300 and AR in this model, unlike the ER/SRC-3/p300 complex (discussed below in Section 5.3) [50]. In concordance with the structural data, the AR NTD is required for the direct AR-p300 interaction based on immunoprecipitation experiments using various AR deletion constructs [50]. Of note, the E1A protein from several species of human adenovirus inhibits AR transactivation function [166]. E1A binds to p300 and the AR, which could interfere with AR-p300 interactions and AR-mediated transcription.

Interestingly, in the cryo-EM structure, SRC-3 does not directly interact with the AR LBD, but instead makes contact with the NTD of one AR monomer [50]. Concordantly, mutation of the three LXXLL motifs in SRC-3 did not affect SRC-3 recruitment to the AR in the presence of androgen [50]. SRC-3 utilizes its bHLH/PAS (basic helix-loop-helix/per-arnt-sim domain) and S/T (serine/threonine rich region) domains to interact with the AR, instead of the SRC-3 RID, which contains the three LXXLL motifs (Figure 4) [50]. SRC-3 is shown to enhance the AR/p300 interaction [50]. Thus, SRC-3 appears to stabilize the AR activation complex [50]. It will be interesting to see if CBP is similarly assembled in the AR activation complex using cryo-EM technology, given that CBP can also interact with the AR [156,167].

### 3.3. p300 Co-Occupies Many Genomic Sites with the AR to Activate AR Target Gene Expression

Early studies showed p300 was a critical AR co-activator using reporter assays. For example, p300 enhances DHT-stimulated AR transcriptional activity at the androgen-responsive MMTV (mouse mammary tumor virus) promoter [76]. Sequestration of p300 with E1A inhibits AR-mediated transactivation of reporter assays driven by the KLK3 promoter [58]. These reporter assays suggest CBP/p300 could be recruited to AR target genes to serve as AR co-activators. Indeed, AR activation with DHT in C4-2B cells results in p300 recruitment and H3K18ac/H3K27ac deposition at the TMPRSS2 enhancer and transcription start site (TSS) [158]. These histone marks, specifically generated by CBP/p300, are characteristic markers for active enhancers and gene transcription [22,23,65,67,69,70]. Likewise, CBP and p300 are recruited to the enhancer and promoter of KLK3, another canonical AR target gene [155,157,168]. Globally, 83% of androgen-induced genes with direct AR binding exhibit overlapping p300 peaks at their promoters or ARE motifs [52].

Consistent with their global genomic co-binding, p300 knockdown interferes with the activation of 52.4% of DHT-induced genes in C4-2B cells (including KLK3 and TMPRSS2) [158]. Another study showed that p300 knockdown affects the expression of 57% of AR target genes [169]. Loss of p300 activity also decreases expression of AR-regulated genes in PTEN-deficient mouse prostate cells and PTEN-deficient PCa cell lines [11]. Furthermore, co-depletion of both CBP and p300 in AR+ PCa cells potently reduces KLK3 and TMPRSS2 expression, with a stronger effect than a single knockdown alone [52]. These observations collectively demonstrate the crucial role for p300 in promoting AR target gene expression.

Where CBP also interacts with the AR and activates AR-mediated transcription [156,167], CBP knockdown seems to have little effect on DHT-stimulated gene expression in C4-2B cells [158]. This suggests that CBP might not be required for AR signaling on a global scale [158]. While CBP and p300 are structurally similar and both interact with the AR, levels of CBP/p300 are thought to be limiting in the cell and their interactome must compete for access to these KATs [12,156,167,170]. Therefore, PCa cells with high levels of p300 and comparatively low levels of CBP may rely more on p300 for AR signaling, and vice versa. Additionally, p300 may be dominant over CBP as an AR interaction partner and CBP may be preferentially recruited to chromatin by other transcription factors.

### 3.4. p300 Stabilizes the AR While Androgen Deprivation Stabilizes CBP/p300

The interaction between CBP/p300 and the AR is important for AR protein stability and response to androgens. In PTEN-deficient models of PCa, CBP/p300 regulates AR protein stability through direct acetylation of K630, K632, and K633 on the AR [11]. Genetic deletion of EP300 reduces AR protein levels in PTEN-deficient mouse prostate tumors and PCa cell lines depleted of PTEN [11]. Mechanistically, PTEN inactivation increases phosphorylation of the AR at Serine 81, which enhances the p300–AR interaction, indicating that Serine 81 phosphorylation is critical for the AR–p300 interaction [11]. In agreement, mutation of S81 to alanine (S81A) greatly attenuates the interaction between the AR and p300 [11]. The AR–p300 interaction facilitates acetylation of K630, K632, and K633 on the AR [11]. Acetylation of these residues inhibits AR polyubiquitination and prevents proteasome-mediated degradation of the AR [11]. Notably, AR protein levels were decreased with p300 knockdown alone (in the absence of PTEN knockdown) in LAPC-4 and 22Rv1 PCa cells [11]. This suggests that p300 may regulate AR protein levels in both PTEN-proficient and -deficient PCa [11].

Acetylation of K630, K632, and K633 by CBP/p300 also regulates AR response to androgens and strengthens AR interaction with p300 [76,171,172]. Mutation of K630, K632, and K633 to alanine attenuates androgen-stimulated AR activation, as measured by AR-responsive reporter assays [76]. Significantly, AR acetylation-mimic mutations (K630Q and K630T) displayed enhanced transcriptional activity, increased interaction with p300, and promoted PCa growth in vitro and in vivo [172]. These results collectively demonstrate that p300 acetylation of the AR is crucial for AR transcriptional activity, protein stability, and PCa growth.

Conversely, androgens and the AR appear to regulate CBP/p300 protein levels [173,174]. Treatment of AR+ PCa cells with high doses (≥1 nM) of R1881, a synthetic androgen, reduces p300 protein levels without affecting mRNA expression [173]. Inhibition of AR activity, with AR antagonists or siRNA, prevents reduction of p300 protein levels with androgen stimulation [173]. This suggests that androgen-stimulated downregulation of p300 expression requires a functional AR [173]. Interestingly, long-term (up to 96 h) androgen treatment also reduces protein levels of other AR co-activators, such as the SRC proteins [168].

In contrast, a low dose (0.1 nM) of R1881 or androgen deprivation appears to stabilize p300 protein [173]. LNCaP-Rf cells, an isogenic cell line established through long-term androgen ablation, have higher p300 levels than parental LNCaP cells and are more dependent on p300 for cell viability and proliferation [173]. Also of note, orchiectomy (surgical removal of the testicles) in mice results in upregulation of p300 and CBP protein levels in LuCaP35 PCa xenografts [175]. Similar to p300, CBP is downregulated by R1881 in LNCaP cells [174]. Unlike p300, CBP is downregulated at the mRNA and protein level with androgen treatment [174]. CBP is reported to be highly expressed in high-grade PCa tumors from patients who failed hormonal therapy [174]. This suggests that CBP may be upregulated as a consequence of endocrine therapy in vivo [174]. These observations suggest that increased CBP/p300 protein levels might be an adaptive response to androgen deprivation to compensate for reduced AR activation in order to sustain cell proliferation and tumor growth [173,174].

### 3.5. CBP/p300 Are AR Co-Activators in the Absence of Androgens

Androgen-independent activation of the AR is a well-known phenomenon and can occur via several different mechanisms, including activation by interleukins [58,60]. For example, interleukin-6 (IL-6) has been shown to activate AR-dependent gene expression in the absence of androgens [58]. Activation of the AR and AR target gene expression by IL-6 requires p300 and its HAT activity [58]. Similar to IL-6, interleukin-4 (IL-4) activates the AR, increases CBP/p300 protein expression, and enhances the interaction of CBP/p300 with the AR at the KLK3 promoter [60]. Therefore, p300 seems crucial for AR transcriptional activity in both the presence and absence of androgens.

Constitutively active AR splice variants (AR-Vs) are another well documented mechanism of androgen-independent activation of AR signaling [134,135,176,177,178,179,180]. More than 20 different AR-Vs have been characterized and can be categorized into three groups: constitutively active, conditionally active, and inactive variants [176]. AR-Vs arise through AR gene rearrangement and alternative mRNA splicing, resulting in truncated forms of the AR that lack the LBD [179,180].

AR-V7 is one of the most highly studied AR-Vs and its expression increases with PCa progression [176,179]. AR-V7 is constitutively active and regulates expression of both canonical AR target genes and a distinct transcriptional program [176,179,181]. Treatment with chemotherapy (taxanes) or AR-signaling inhibitors (enzalutamide or abiraterone) appear to lead to increased AR-V7 expression [182], which, nonetheless, does not correlate with patient survival. As discussed above, p300 interacts with the AR through the AR NTD in the AR/SRC-3/p300 cryo-EM structure [50]. Most AR-Vs, including AR-V7, retain the NTD and thus are capable of binding to p300 along with SRC proteins such as SRC-1 and SRC-3 [50,176,183]. Based on these observations, it seems clear that p300 is an important co-activator for AR-Vs.

## 4. CBP/p300 Represent Rational Drug Targets in AR+ PCa

### 4.1. p300 Is Critical for PCa Tumor Growth

Increasing evidence shows that p300 promotes PCa tumorigenesis, tumor progression, and treatment resistance [11,184,185,186]. Notably, p300 is upregulated in tumor samples from PCa patients treated with docetaxel, a commonly used microtubule inhibitor for treating CRPC [186]. Docetaxel-resistant PCa cell lines express higher levels of p300 and p300 knockdown impairs clonogenic growth of these cell lines, suggesting that p300 may play a role in resistance to docetaxel [186]. As described in Section 3.3, PTEN inactivation was shown to increase AR phosphorylation, which enhances the interaction between AR and p300, AR acetylation and stability [11]. Consistently, genetic EP300 ablation attenuates AR expression and inhibits tumor formation in a mouse model of PTEN deletion-induced tumorigenesis in vivo [11].

A potential role for CBP in PCa appears more complex. Both high expression and downregulation of CBP in PCa have been reported [10,174,187]. In tumor samples, CBP expression positively correlates with PTEN expression [10]. Interestingly, concomitant deletion of both copies of CREBBP and one allele of PTEN in mouse prostatic epithelial cells promotes cell proliferation and tumorigenesis in vivo [10]. At the molecular level, CBP and PTEN loss results in decreased H3K27ac levels, but increased levels of EZH2 and H3K27me3, which may cause reduced expression of growth inhibitory genes [10]. In one study, p300 knockdown, but not CBP knockdown, reduces proliferation, inhibits invasiveness, and increases apoptosis in LNCaP cells [63]. Notably, regardless of AR expression status, the growth of some PCa cell lines is not affected by CBP knockdown, while the survival of other cell lines is pronouncedly impaired [28,52]. Remarkably, in all cell lines studied, co-depletion of CBP and p300 has stronger growth inhibitory effects than either depletion alone [28,52]. This is consistent with the hypothesis that p300 has a dominant role in PCa tumorigenesis, while the oncogenic function of CBP may become more important when p300 is downregulated. Collectively, these studies provide evidence that both p300 and CBP can promote PCa tumor growth and represent rational drug targets in PCa treatment.

### 4.2. CBP/p300 BD Inhibitors Show Potent Anti-Proliferative Effects in Preclinical PCa Studies

GNE-049 is a potent CBP/p300 BD inhibitor with an IC_50_ of 1.1 nM for CBP and 2.3 nM for p300 in a biochemical BD-binding assay [52]. This compound is highly selective for CBP/p300 over other BD proteins, such as BRD4 (BRD4 IC_50_ of 4240 nM) [52]. GNE-049 reduces proliferation of AR+ PCa cell lines in both 2D and 3D growth assays [52]. In contrast, the viability of AR— PCa PC3 and DU145 cell lines was not impacted by GNE-049 [52]. GNE-049 reduces expression of AR target genes KLK3 and TMPRSS2 in AR+ PCa cells to a greater degree than enzalutamide [52]. Importantly, GNE-049 impairs AR signaling in 22Rv1 cells, which express the AR-V7 variant that lacks the LBD [52]. Therefore, AR signaling in 22Rv1 cells is not inhibited by enzalutamide (an AR LBD antagonist) but is sensitive to CBP/p300 inhibition [52]. RNA-seq analysis of AR+ PCa cells demonstrates that GNE-049 suppresses the expression of AR target genes [52]. Significantly, GNE-049 treatment reduces androgen-stimulated growth in AR+ PCa cells [52]. In vivo, GNE-049 potently suppresses the growth of AR+ patient-derived xenograft (PDX) PCa models [52].

Mechanistically, Jin et al. proposed that GNE-049 reduces AR signaling and PCa growth by blocking CBP/p300 co-activator function [52]. Consistent with this model, GNE-049 reduces H3K27ac at the AR/p300 genomic binding sites [52]. Interestingly, GNE-049 does not prevent recruitment of the AR or p300 to chromatin or inhibit the interaction between the AR and p300 [52]. Therefore, GNE-049 does not disrupt the formation of the AR co-activator complex, but instead functions to inhibit CBP/p300 as a co-activator [52]. As described in Section 2.2, GNE-049 seems to have a stronger inhibitory effect on H3K27ac at enhancers, in comparison to promoters [23]. These findings suggest that GNE-049 treatment may selectively block the activity of ARE-containing enhancers, thereby suppressing the expression of oncogenes. Of note, GNE-781, a structural analog of GNE-049, was developed to address an adverse effect on the central nervous system by GNE-049 in vivo [91,94]. GNE-781 maintains the potency and selectivity of GNE-049 but is less brain-penetrant [91,94]. Therefore, GNE-781 may be a safer alternative for in vivo studies.

Additional CBP/p300 BD inhibitors have been characterized and show efficacy in PCa treatment. NEO2734 is a broad BD inhibitor with single nanomolar K_d_ values for BRD2, BRD3, BRD4, and BRDT, and K_d_ values of 19 and 31 nM for CBP and p300, respectively [188]. NEO2734 inhibits growth of LNCaP-FGC, 22Rv1, and VCaP PCa cells with IC_50_ values of 0.24, 0.61, and 0.17 µM, respectively [189]. NEO2734 also reduces in vivo tumor growth of VCaP xenografts [189]. Yan et al. demonstrated that NEO2734 has pre-clinical efficacy in SPOP wild-type (WT) and mutant PCa [110]. SPOP is the most frequently mutated gene in primary PCa (10–15% of cases) and is a substrate receptor subunit of the Cullin 3 E3 ubiquitin ligase complex [110,190,191]. SPOP promotes proteasomal degradation of the AR [192], AR co-activators [193,194,195,196], and other proteins, such as ERG [197] and MYC [198]. SPOP mutations impair AR degradation, consequently, AR signaling is highly active in SPOP mutant PCa [192]. NEO2734 reduces AR protein levels and expression of AR target genes in both SPOP WT and mutant AR+ PCa PDX models [110]. NEO2734 inhibits the growth of SPOP WT and mutant PCa organoids, PDXs in vivo, and AR+ SPOP mutant C4-2 cells [110].

Y08197 is another CBP/p300 BD-selective inhibitor with an IC_50_ of approximately 100 nM for the CBP BD [111]. Y08197 inhibits the expression of AR target genes KLK3 and TMPRSS2 in AR+ PCa cell lines, including 22Rv1 cells [111]. Significantly, Y08197 reduces mRNA expression of AR-full length (AR-FL) in LNCaP cells and AR-V7 in 22Rv1 cells [111]. Y08197 also reduced growth of AR+ PCa cell lines [111]. CPI-637 is another CBP/p300 BD-selective inhibitor with >700-fold selectivity over the BET family of BDs [108]. This inhibitor was also effective in inhibiting AR signaling and cell proliferation in PCa cells [110,111].

Collectively, these preclinical studies provide compelling evidence that targeting the CBP/p300 BD represents a promising strategy for treating PCa. Notably, a Phase I/IIa clinical trial (ClinicalTrials.gov Identifier: NCT03568656) was initiated for CBP/p300 BD inhibitor CCS1477 for treatment of metastatic PCa [94,199]. This inhibitor shows both high affinity and selectivity for the CBP/p300 BD and downregulates AR signaling in AR+ PCa cells [199]. Further clinical studies of CBP/p300 BD inhibitors for treating PCa are anticipated.

### 4.3. The CBP/p300 HAT Inhibitor A-485 Is Effective against PCa In Vitro and In Vivo

Earlier studies have shown that pharmacologic inhibition of CBP/p300 HAT activity represents an effective strategy for treating PCa. C646, a CBP/p300 HAT inhibitor identified through a structure-based virtual screen [83], increases apoptosis and reduces PSA secretion in AR+ PCa cells [63]. This suggests C646 acts through inhibition of the AR pathway [63]. As discussed above, A-485 is currently the most potent and selective catalytic inhibitor of CBP/p300 [28]. In PC3 cells, A-485 inhibits H3K27ac with an EC_50_ of 73 nM [28]. Interestingly, A-485 potently inhibits proliferation of AR+ cells, but not AR— cells in vitro, and reduces in vivo tumor growth of AR+ xenografts [28]. Thus, the anti-proliferative effects of A-485 may be attributed to its ability to inhibit AR signaling.

Indeed, A-485 reduces expression of known AR targets, such as KLK3, TMPRSS2, and SLC45A3 [28]. On a global scale, A-485 antagonizes the expression of androgen-upregulated genes (but has little modulation of androgen-repressed genes) both in LNCaP-FGC cells and 22Rv1 cells [28]. Mechanistically, A-485 does not reduce AR recruitment to AR target gene KLK3; however, A-485 does reduce H3K27ac levels at the ARE in the KLK3 enhancer [28]. Therefore, A-485, similar to GNE-049, suppresses gene expression by inhibiting the co-activator function of CBP/p300 instead of affecting AR complex formation and the chromatin-binding of the AR [28].

Interestingly, A-485rs (a mixture of two A-485 diastereomers) synergizes with CBP/p300 BD inhibitor I-CBP112 to dramatically reduce p300 occupancy genome-wide [120]. This results in a synergistic inhibitory effect on PCa proliferation and a strong inhibitory effect on gene expression in LNCaP cells [120]. These observations provide a preclinical proof-of-concept that agents inhibiting the CBP/p300 HAT function may be effective for treating PCa and that a combination of CBP/p300 HAT and BD inhibitors may be synergistic in blocking CBP/p300 function [28].

## 5. The Role of CBP/p300 in ER Signaling in ER+ BC

### 5.1. BC Is a Diverse Disease with Hormone-Dependent and Independent Subtypes

BC is the leading cause of cancer burden in women and is diagnosed in over 1 million women worldwide each year. BC affects 1 in 20 women globally and as many as 1 in 8 in Western countries. BC is a heterogeneous disease with four main molecular subtypes: Luminal A, Luminal B, Her2-overexpression (HER2-enriched), and basal-like [40]. ERα, or simply ER (encoded by the ESR1 gene), progesterone receptor (PR, encoded by the PGR gene), and HER2 (encoded by the ERBB2 gene) are diagnostic markers for the four major subtypes. Luminal A is ER+ and/or PR+, but HER2-; Luminal B is ER+ and/or PR+, plus HER2+; HER2-enriched is ER—but with overexpressed HER2; and basal-like is negative for ER, PR, and HER2. The basal-like subtype is also commonly known as triple-negative BC (TNBC), among which basal-like tumors are the major components. According to the current survey by the American Cancer Society, approximately 71% of BC cases in the United States are luminal A, 12% luminal B, 12% TNBC, and 5% HER2-enriched. The proportion of TNBC cases is higher for patients younger than 50 years of age (15%) and of African American descent (23%). About 20–30% of invasive BCs progress to recurrent or metastatic disease and ~90% of BC deaths are due to metastatic cancer in distant organs, such as the brain, liver and lungs. Although more BC patients survive the disease, over 450,000 patients die of this disease annually. In 2018, it was estimated that 40,920 BC deaths occurred in the United States alone [200]. This makes BC the second leading cause of cancer-related mortality in women in the United States [200].

ER+ BC constitutes approximately 60–80% of BC cases (depending on the study) and most BC deaths occur in women with ER+ BC (Figure 2A) [201,202]. ER signaling is the major driver of growth in ER+ Luminal A BC and has also made major contributions to growth in ER+ Luminal B BC [40,47,203,204]. As such, ER+ BC is generally treated with hormonal therapies that block estrogen signaling, such as aromatase inhibitors and selective ER modulators [205,206]. HER2-targed therapies, such as trastuzumab and lapatinib, are used for treating patients with HER2-enriched BC [207]. Unfortunately, no targeted therapies are available for TNBC, which is commonly treated with chemotherapy [208,209]. Interestingly, recent studies have characterized a subgroup of TNBC that is dependent on AR signaling, similar to PCa [210]. Whether targeting AR signaling is effective for treating AR+ TNBC is currently being explored [210,211].

While there is clear evidence that current BC treatments have prolonged patient survival [212], de novo and acquired resistance to current treatments is common [207,209,213,214]. Therefore, new therapies are needed to improve clinical outcomes for many BC patients.

### 5.2. ER Signaling Drives Tumor Growth of Luminal BC Subtypes and Is the Major Therapeutic Target

The full-length ER is a 66 kDa steroid hormone nuclear receptor that shares structural similarity to the AR and other nuclear receptors [215]. Like the AR, the ER has an LBD, a DBD, and an N-terminal intrinsically disordered region acting as a transcriptional activation domain [215]. In the absence of ligand, the ER is sequestered in the cytoplasm by heat shock proteins [216,217]. When the ER binds 17β-Estradiol (E2), the main circulating estrogen hormone in the female body, it undergoes homodimerization and translocates to the nucleus where it binds estrogen response elements (EREs) to activate or repress transcription (Figure 3B) [40,215,216]. Similar to the AR, other transcription factors, such as the pioneering factor FOXA1, can enhance ER recruitment to ERE sites and impact ER complex formation (Figure 3B) [125].

ER signaling is pro-growth through activation of oncogenes, such as MYC and CCND1, to promote cell cycle progression [45,47,204,218]. The ER can also be localized to the plasma membrane and can regulate cellular biology independent of its function as a transcription factor [217,219]. For example, membrane-localized ER mediates rapid activation of kinase signaling pathways [217,219]. This review focuses on the transcriptional activity of the ER; therefore, we refer readers to this in-depth review on membrane-bound ER by Acconcia and Marino [217].

The ER has a dominant oncogenic role in ER+ BC, and ER-targeted therapeutics are the mainstay therapy for ER+ BC patients. Tamoxifen is a selective inhibitor of ER function in the breast and has been the standard therapy for ER+ BC for decades [56,220]. Unfortunately, the use of tamoxifen in the clinic is complicated by de novo or acquired resistance [56,220]. Approximately 20–30% of tumors are refractory to tamoxifen treatment and new therapies are needed to address this clinical challenge [56,203,220]. Tamoxifen-resistant tumors remain sensitive to other inhibitors of ER function, which implies that these tumors still rely on ER signaling for growth [56]. As with targeting the AR in PCa, targeting ER critical co-activators is an emerging strategy for treating ER+ BC [221].

### 5.3. CBP/p300 Are Components of the ER Activation Complex

The ER relies on transcriptional co-activators to upregulate its target genes via the well characterized ER activation complex (ER/SRC-3/p300) [51,163,215,222,223,224,225]. As noted above, upon binding of ligand, the ER dimerizes, translocates to the nucleus, and binds ERE sequences (Figure 3B) [40,215,216]. Each ER monomer independently recruits one SRC-3 protein through direct binding of SRC-3 to the ER LBD [51]. p300 is then assembled to the complex by interacting with the two SRC-3 proteins (Figure 3B) [51]. In this model of complex formation, SRC-3 acts as an adaptor to bridge the interaction between the ER and p300 [51]. Other co-activators may also be involved in the ER complex [226]. For example, an additional cryo-EM study by Yi et al. revealed that CARM1, an arginine methyltransferase, is sequentially recruited to the ER/SRC-3/p300 complex and replaces one SRC-3 protein in the complex [225], analogous to the CARM1 recruitment to the AR target genes by androgen-activated AR [165], as discussed above. CARM1 interacts with p300 via its N-terminal domain and enhances p300 HAT activity toward H3K18 [225]. CARM1 in the complex methylates H3R17 at ER target genes [225]. CBP is also detected in the ER transcription complex [227]. However, CBP appears to be recruited at a later time point than p300, which is rapidly recruited upon estrogen stimulation [227].

The SRC proteins, including SRC-3, directly interact with the LBD of the ER via an LXXLL motif in their RID (receptor interaction domain, Figure 4) [51,161,163]. The LXXLL motif is a signature motif utilized by co-regulators to directly bind steroid nuclear receptors [161]. Notably, the AF-1 domain of the ER (within the ER N-terminal region) also helps recruit SRC-3 to the complex [51]. The SRC’s CID (CBP/p300-interaction domain) then binds p300’s SID (SRC-interacting domain) for tethering p300 to the ER activation complex (Figure 4) [51,162,163]. Interestingly, the cryo-EM structure of the ER/SRC-3/p300 complex reveals that p300 has multiple interactions with SRC-3, which are mediated by the p300 NRID, C/H1, KIX, and SID domains [51,225]. This suggests that the interaction between p300 and SRC-3 is more complex than a simple binary interaction between SRC’s CID and p300’s SID. However, deletion of the SRC-3 CID blocks p300 interaction with the ER and prevents SRC-3 from acting as an ER co-activator [51]. Additionally, dominant-negative polypeptides of the SRC-2 RID and p300 SID inhibit ER transcriptional activity in vitro by blocking complex formation [163]. Therefore, the SRC-3 CID and p300 SID interaction is critical for complex formation and ER activity.

Similar to the SRC proteins, CBP/p300 also possess LXXLL motifs in their N-terminus for direct binding to the ER, which is observed between CBP and the ER [161,170]. However, direct interaction between CBP/p300 and the ER appears weak and binding of p300 to the ER/DNA complex is not detected in the absence of SRC-3. [51,162]. Collectively, these results demonstrate that the SRC proteins are crucial for recruiting CBP/p300 to the ER complex. ER also interacts with several co-repressor complexes as well as chromatin remodelers, which was reviewed recently [226].

### 5.4. CBP/p300 Co-Bind with ER at Many Genomic Sites

Genome-wide ChIP-seq experiments in ER+ BC MCF-7 cells show that approximately 30% of all ER binding sites are shared by p300 [51]. Conversely, approximately 56% of all p300 binding sites are shared with the ER, demonstrating that the ER is a prominent interaction partner for p300 in ER+ BC [51]. These results imply that p300 is important for the expression of many ER targets. Indeed, p300 binding is correlated with the transcriptional regulation of estrogen target genes [228]. MCF-7 cells treated with E2 display a redistribution of p300 from regions not bound by the ER to regions co-occupied by ER/p300 [228]. The genomic sites that gain p300 binding are enriched near genes transcriptionally upregulated by estrogen treatment, whereas genomic sites that lose p300 binding are enriched near genes transcriptionally downregulated by estrogen [228]. Other studies also demonstrate a redistribution of p300 due to hormone stimulation in ER+ BC [53,229].

Similar to p300, CBP is recruited to ER binding sites upon estrogen stimulation. In one study, approximately 38% of CBP binding sites were shared by the ER [53]. Further analysis revealed that approximately 69% of ER binding sites are co-occupied by both CBP and p300 [53]. Differential binding sites with the ER were also identified for CBP and p300 [53]. This suggests that a subset of ER binding sites may be specifically regulated by p300 or CBP, but not both [53]. Motif analyses demonstrate that CBP/p300 binding sites near the TSS of E2-regulated genes have both distinct and common motif enrichments between CBP and p300 [53]. For example, p300 binding sites are enriched for forkhead motifs, while CBP binding sites are not [53]. These genomic studies further support the critical roles of CBP and p300 in ER signaling. Additionally, they suggest that CBP and p300 may have both redundant and distinct roles in regulating ER target gene expression.

### 5.5. The HAT Activity of CBP/p300 Is Critical to ER Signaling

Early studies clearly demonstrated that CBP and p300 act along with SRC proteins to co-activate ER-mediated transcription [222,230]. These findings raised the question of whether CBP/p300 HAT activity is critical for co-activation of ER signaling. Several studies showed that CBP/p300 HAT activity is required for ER-mediated transcription and that CBP/p300 binding to the ER complex enhances its HAT activity toward H3 [51,163,225]. For example, the ER/SRC/p300 complex and estrogen stimulation appear to promote efficient acetylation of an ERE-containing chromatin template [163]. Furthermore, pharmacological inhibition of p300’s HAT activity blocks estrogen-stimulated ER target gene expression [118].

Interestingly, the formation of the ER activation complex also promotes CBP/p300-mediated acetylation of the ER at several lysine residues, which impacts ER function [80]. For example, K266 and K268 in the ER DBD are acetylated by p300 [80]. ER K266/268Q mutants that mimic acetylated ER exhibit increased DNA-binding activity and transcriptional activity [80]. These results indicate that acetylation of the ER is an important regulatory post-translational modification [80]. In addition, p300 acetylation of the ER promotes ER protein stability, potentially through reducing ubiquitination and proteasomal degradation of the receptor [54].

A recent study reported that H3K27ac at the TSS of the ESR1 gene is important for its expression [24]. Mechanistically, the scaffold subunit (AFF4) of the super elongation complex (SEC) interacts with H3K27ac at the TSS to promote ESR1 expression [24]. As CBP/p300 specifically acetylate H3K27, this study provides evidence that CBP/p300 are important for ER expression at the transcriptional level, in addition to their effects on the ER at the post-translational level.

### 5.6. p300 Contributes to ER-Mediated Transcriptional Repression

Estrogen treatment both activates and represses a large number of genes. The ER target gene network is involved in apoptosis, cell cycle regulation, cell adhesion, signal transduction, and other functional categories [231,232]. Notably, estrogen stimulates a greater proportion of genes than it represses in the cell cycle and nucleotide processing categories [232]. In comparison, estrogen represses a greater proportion of genes than it activates in the cell adhesion/extracellular matrix and signal transduction categories [232]. Cyclin G2 (CCNG2), a negative regulator of the cell cycle, is an example gene that is directly repressed by the ER [233]. As detailed above, much work has been done to elucidate the mechanisms by which the ER activates its target genes. However, the mechanisms behind ER-mediated gene repression are less understood.

CBP/p300, in addition to their roles as co-activators for many signaling pathways, also play a role in transcriptional repression [234,235,236,237]. Stossi et al. reported that treatment of MCF-7 cells with estrogen resulted in p300 recruitment to estrogen-repressed genes (e.g., CCNG2, MMD, and SMAD6) and that p300 knockdown blocked their transcriptional repression [238]. Mechanistically, p300 promotes ER-mediated repression through recruitment of CtBP1, a known repressor of p300 HAT activity via direct binding to the p300 BD [238,239]. The recruitment of CtBP1 to repressed ER target genes suggests that p300 may not be acting as a functional HAT at these genes [238]. Given the recent findings that the BD of p300 regulates its HAT activity [22,23], the binding of CtBP1 to the BD of p300 could play a role in blocking its HAT function. In line with this, CtBP1 recruitment at these ER-repressed sites resulted in lower levels of histone acetylation [238]. This may be due to decreased p300 activity and the recruitment of histone deacetylase (HDAC) enzymes to the chromatin by CtBP1 [238,240]. The CtBP proteins (CtBP1 and CtBP2) are upregulated in cancer, including BC, and are emerging important regulators of BC biology [240,241,242].

Other mechanisms of ER-mediated transcriptional repression have also been proposed. Guertin et al. demonstrated that activated ER redistributes p300 binding from estrogen-repressed genes to estrogen-activated genes, resulting in repression of certain genes [229]. For example, estrogen downregulates MAPK9 expression and reduces p300 binding at the MAPK9 enhancer [229]. In comparison, estrogen induces TFF1 expression and upregulates p300 binding at a nearby ER-bound enhancer [229]. As detailed above, other groups have reported redistribution of p300 upon E2 stimulation in ER+ BC cells and that loss of p300 binding correlates with decreased transcription [53,228].

Interestingly, Zwart et al. demonstrated that E2-repressed genes are pre-bound by CBP and p300 prior to ER recruitment and their binding is independent of estrogen treatment [53]. These findings suggest that the timing of p300 binding could impact the outcome of gene regulation. Whether CtBP1 and other known co-repressors are associated with the pre-bound p300 in this scenario is an intriguing mechanistic question. Precisely how CBP/p300 is involved in ER transcriptional repression remains to be established.

### 5.7. CBP/p300 Have Enhanced Interactions with Constitutively Active ER Mutants and p300 Is Essential for ER Mutant BC Cell Growth

As stated above, resistance to tamoxifen is common in ER+ BC [56,203,220]. The mechanisms that drive tamoxifen resistance have been extensively studied and the emergence of constitutively active ER mutants following therapy is a major resistance mechanism [59,61,62,204,213,214,220,243,244,245,246,247,248,249,250,251]. Mutations in the ESR1 gene were found in 10% of patients and the most frequently occurring mutations were D538G (36%) and Y537S (14%) in the LBD of the ER [247]. Another study found ESR1 mutations to be more common, even as high as 32% in metastatic BC that had been previously treated with hormone therapy [61]. Interestingly, 3% of primary tumors analyzed from the BOLERO-2 clinical trial exhibited ER LBD mutations [61]. This is in contrast to other studies that reported ER mutants are not present in primary treatment-naïve tumors [243,245]. Therefore, the occurrence of ER LBD mutations in primary tumors should be examined in further studies.

Characterization of the D538G and Y537S mutants is an ongoing area of research. Recent evidence shows that these mutants are constitutively active in the absence of E2, can drive E2-independent growth in ER+ BC models, and confer drug resistance to standard endocrine therapies, such as tamoxifen or fulvestrant [59,61,245,247,249]. The constitutive activity of the D538G and Y537S mutants is due to a stabilized agonist state that exhibits increased interactions with ER co-activators, such as SRC-3 and CBP/p300 [59,61,244,247].

MCF-7 cells expressing the ER Y537S mutant gain a significant number of H3K27ac sites in hormone-depleted conditions vs. WT ER cells stimulated with E2 [59]. Importantly, these H3K27ac sites were enriched for the ERE motif [59]. Greater than 30% of known super enhancers in the MCF-7 Y537S cells overlap with mutant gained ER binding sites [59]. Super enhancers consist of large clusters of enhancers and are known to regulate expression of oncogenes, such as MYC [66,68]. These findings imply that CBP/p300 HAT activity may be critically important for ER mutant function because CBP/p300 specifically acetylate H3K27 in enhancers and super enhancers [23,65,66,68,252]. Indeed, p300 was found to be essential for cell growth in T-47D cells expressing ER Y537S using a CRISPR gene knockout viability screen [59]. These observations suggest that targeting CBP/p300 in these clinically challenging mutant tumors may be a valid therapeutic strategy, given that CBP/p300 inhibitors potently downregulate H3K27ac at enhancers.

## 6. CBP/p300 Represent Rational Drug Targets in BC

### 6.1. CBP/p300 Inhibitors Suppress ER Signaling and ER+ BC Growth

Given the well documented roles for CBP/p300 as critical ER co-activators, pharmacological inhibition of CBP/p300 may represent a valid therapeutic strategy for treating ER+ BC. In a high throughput screen, L002 was identified as a HAT inhibitor that targets CBP/p300 [253]. L002 is cytotoxic in ER+ BC cells and potently reduces MCF-7 colony formation [253]. CBP/p300 HAT inhibitor C646 is also reported to block E2-induced growth and expression of ER target genes in MCF-7 cells [118]. In the ER activation complex, the p300 BD is free to engage substrates [51]. This implies that p300 BD inhibitors may also affect p300 activity at ER target genes [51]. In support of this, Murakami et al. demonstrated that SGC-CBP30, a CBP/p300 BD inhibitor, reduces p300 recruitment to ER binding sites, suppresses E2-activation of ER target gene GREB1, and blocks E2-induced growth of MCF-7 cells [118]. CBP/p300 BD inhibitor Y08197 also displays efficacy in inhibiting MCF-7 proliferation with an IC_50_ of 10.61 µM [111]. These studies provide a proof-of-principle for targeting CBP/p300 in ER+ BC with a small molecule inhibitor. The recently developed CBP/p300 HAT (A-485) and BD (GNE-049) inhibitors are significantly more potent and selective [28,52]. In a recent study, A-485 and GNE-049 were shown to impair ER-mediated MYC and CCND1 expression in ER+ BC cell lines. On a genomic scale, A-485 markedly reduces the levels of H3K27ac on enhancers. Both A-485 and GNE-049 potently inhibit proliferation of ER+ BC cells. A-485 inhibits MCF-7 cell growth apparently through inducing senescence [55].

### 6.2. Targeting CBP/p300 to Inhibit AR Signaling as a Potential Therapeutic Strategy in AR+ BC

Whereas the AR is a known driver of PCa growth, the AR is also expressed in most cases of BC (60 to 80%). Depending on the definition of AR positivity, about 70 to 90% of ER+ and 10 to 35% of TNBC express the AR [254]. In ER+ BC, the expression of the AR appears to confer a favorable prognosis [255], which was ascribed to an inhibitory role of the AR in ER signaling [254]. Co-expression of HER2 and the AR might also be associated with less aggressive tumors [254]. However, the AR was shown to mediate a gene expression program underlying epithelial to mesenchymal transition and metastasis through an AR–LSD1 (lysine specific demethylase 1) interaction [254]. Furthermore, high nuclear levels of the AR appear to contribute to resistance to hormone therapy and confer a worse prognosis [211,254,255].

The AR is becoming increasingly recognized as a potential therapeutic target in TNBC, which was previously considered to be hormone-independent. Although TNBC is collectively classified as a major BC subtype, it is highly heterogeneous and can be further divided into six stable subgroups [210]. One subgroup, known as the luminal androgen receptor (LAR) subgroup, represents 10–15% of all TNBC cases (Figure 2A) [210,256]. The heterogeneity of TNBC and the existence of the LAR subgroup were confirmed by subsequent clinical studies [257]. LAR TNBC is characterized by high levels of the AR and expression of AR target genes, such as FKBP5 [210]. Similar to AR+ PCa cell lines, LAR cell lines are sensitive to AR antagonists and siRNA knockdown of the AR reduces their clonogenic growth [210]. Androgen treatment also enhances invasion of LAR cell lines, which is dependent on AR function [258]. Thus, the AR appears to be an oncogenic driver for LAR TNBC.

LAR TNBC is not the only subgroup to express or rely on the AR for tumor progression [210,259,260]. Indeed, recent evidence shows that cell lines from the mesenchymal-like (MSL), basal-like 1 (BL1), and basal-like 2 (BL2) TNBC subgroups also express the AR [210,260]. In LAR and other TNBC subgroups, the AR appears to protect against anoikis (detachment-induced cell death) and promote a cancer stem cell-like (CSC-like) population [259]. Pretreatment of SUM159PT cells (MSL subgroup) with enzalutamide reduces tumor initiation frequency in vivo [259]. Androgen stimulation in SUM159PT cells increases baseline proliferation, which is reversed by AR knockdown or enzalutamide [260]. Additionally, enzalutamide reduces clonogenic growth, increases apoptosis in vitro, and decreases viability (increased apoptosis/necrosis) of SUM159PT-derived tumors in vivo [260]. These preclinical data provide a rationale for clinical testing of antiandrogens, such as enzalutamide, for treating AR+ TNBC. A phase II trial using AR antagonist bicalutamide in ER-/PR-/AR+ metastatic BC showed a 19% clinical benefit rate (CBR), providing a clinical proof-of-principle for targeting the AR in TNBC [261].

Interestingly, LAR TNBC cell lines are sensitive to CBP/p300 inhibition. The MBA-MB-453 LAR cell line was sensitive to the CBP/p300 HAT inhibitor C646 and BD inhibitors SGC-CBP30 and I-CBP112 [262]. Mutation of the CBP/p300 BD using CRISPR genome editing in MBA-MB-453 cells also impaired proliferation, confirming the specificity of the CBP/p300 inhibitors in this model [262]. Chemical inhibition of the CBP/p300 BD reduced AR transactivation activity, H3K27ac at AR binding sites, and the expression of MYC [262]. These results suggest a potentially critical role for CBP/p300 in AR signaling in LAR TNBC. This work suggests therapeutic targeting of CBP/p300 is a promising strategy for treating AR+ TNBC.

### 6.3. CBP/p300 Promotes Sex Hormone-Independent Oncogenic Signaling Pathways in BC

CBP/p300 also have significant roles in other oncogenic pathways that operate in ER+ and other BC subtypes. Earlier studies have shown that acetylation of STAT3 by p300 is required for its tyrosine phosphorylation and thus its nuclear translocation and activation [263,264]. Increased STAT3 acetylation is observed in human cancer specimens [265]. Genetic ablation of site-specific acetylation of STAT3 reactivates the expression of tumor suppressor genes and promotes chemosensitization [265]. Suppression of STAT3 acetylation leads to inhibition of tumor growth in a mouse TNBC xenograft model [265].

Additionally, there is extensive literature documenting the important roles of CBP/p300 in the Wnt/ß-catenin pathway. These KATs physically interact with ß-catenin and serve as co-activators for ß-catenin/TCF-mediated gene expression [266,267]. CBP/p300 acetylates ß-catenin [268,269,270]. Acetylated ß-catenin is more stable and has higher binding affinity to TCF [268] and to CBP/p300 [266]. Wnt signaling and glucose cooperatively promote ß-catenin acetylation and nuclear translocation, resulting in the expression of genes involved in insulin signaling and cell proliferation [270]. This finding provides a possible link between high serum glucose level and cancer. The activation of the Wnt-ß-catenin pathway is critical for metastatic progression of the HER2-enriched BC subtype [271] and the basal/TNBC subtype [272,273]. Notably, CBP was shown to be critical to the Wnt-ß-catenin signaling underlying the metastatic progression of both HER2-enriched BC and TNBC [271]. ICG-001, a small molecule compound that specifically disrupts the binding of ß-catenin to CBP [274], suppresses cancer progression [271,272].

CBP/p300 bind specifically to HIF-1α and are critical co-activators for HIF-1α-mediated gene expression under hypoxia conditions, which underlies tumor angiogenesis, invasion, metastasis, and resistance to therapy [275,276,277]. Notably, the lysine acetyltransferase PCAF (p300/CBP-associated factor, also known as KAT2B) acetylates K674 of HIF-1α, which enhances its interaction with p300 at the promoters of HIF-1α target genes [278]. An important role for p300 in the HIF-1α pathway has also been documented in clinical BC tumor samples [279]. Collectively, it is evident that CBP/p300 promotes multiple hormone-independent oncogenic signaling pathways that contribute to BC progression. These observations suggest agents that block CBP/p300 activity, such as the HAT and BD inhibitors, may be effective for treating BC.

Of note, several recent studies demonstrated important regulatory roles of histone variant H3.3 in regulating CBP/p300 activity [280,281,282]. H3.3 differs from the canonical histone variant H3.1 by several residues including residue 31, which is a serine in H3.3 and an alanine in H3.1. Interestingly, H3.3S31 phosphorylation by the kinase Chk1 stimulates the histone acetyltransferase activity of CBP/p300, increasing levels of H3K27ac in mouse embryonic stem cells [281]. In activated macrophages, the kinase IKKα phosphorylates H3.3S31, promoting the recruitment of the H3K36 methyltransferase SETD2 [280]. H3K36me3 inhibits the catalytic function of the H3K27 methyltransferase complex PRC2 [283]. As PRC2 opposes CBP/p300 catalytic activity, SETD2 therefore indirectly promotes CBP/p300-mediated H3K27 acetylation [284]. H3F3A and H3F3B (both encoding H3.3) mRNA levels are highly upregulated in BC and other cancers [285]. Therefore, elevated H3.3 may promote the oncogenic function of CBP/p300. It will be interesting to establish the functional link between H3.3 and CBP/p300 in hormone-dependent and -independent cancers in future studies.

## 7. Conclusions

CBP/p300 are critical co-regulators for diverse transcription factors that underlie numerous signaling pathways in normal and cancerous tissues. There is substantial preclinical evidence that CBP/p300 are valid therapeutic targets in PCa, but work remains to be done to determine the precise mechanisms of action of CBP/p300 inhibitors. For example, the CBP/p300 HAT inhibitor A-485 robustly downregulates signaling pathways in PCa that are weakly regulated or unaffected by AR antagonists [28]. These pathways include MYC, ERK, WNT1, WNT3A, and HIF1A signaling [28]. As discussed above, CBP/p300 are critically involved in the WNT/β-catenin and HIF1α pathways [266,267,275]. Thus, CBP/p300 inhibitors may have broader effects on signaling pathways than AR antagonists and the anti-proliferative effects of novel CBP/p300 inhibitors in PCa may not be fully attributable to their inhibition of AR. Further investigation into the broader impact of CBP/p300 inhibitors on signaling pathways will provide important insights into the mechanisms of action and pharmacodynamic effects of these agents.

There is a compelling rationale for using CBP/p300 inhibitors to treat other hormone receptor-dependent cancers. ER+ BC subtypes, as well as ER-negative BC subtypes expressing the AR, represent largely unexplored opportunities to apply these novel CBP/p300 inhibitors. Rigorous validation of the specificity and mechanisms of action of these compounds will be important in any new cancer model. A combination of small molecule inhibitors with genetic knockdown studies of CBP/p300 will be key to identifying and validating ER target genes that rely on CBP/p300 for their expression. It will be important to identify ER target genes whose expression relies on CBP/p300 in tumor cells expressing WT ER or its mutants that confer resistance to hormone therapy. Such CBP/p300-regulated ER target genes may serve as biomarkers for assessing treatment response to these new compounds.

As a future perspective, exploration of CBP/p300 in immune modulations and cancer immunotherapy may lead to new therapeutic strategies. Indeed, p300 is implicated in promoting the expression of CD274 encoding programmed death-ligand 1 (PD-L1), which may confer resistance to cancer immunotherapy in PCa. A-485 inhibits CD274 expression in PCa cell lines [86]. When combined with an anti-PD-L1 antibody, A-485 markedly suppresses tumor growth in vivo [86]. The potential enhancement of immunotherapy efficacy by CBP/p300 inhibitors will spur further investigation in this emerging front. Novel approaches, such as a combination of GNE-049 with immuno-checkpoint blockade, merit further investigation for treating BC and PCa that are largely refractory to immunotherapy. Notably, a recent study showed that p300 acetylates PD-L1, resulting in reduced nuclear localization of PD-L1 and expression of genes that affect antitumor efficacy of anti-PD-L1 antibody [286]. HDAC2 deacetylates PD-L1 to promote PD-L1 nuclear localization and resistance to anti-PD-L1 treatment. A combination of anti-PD-L1 with an HDAC2 inhibitor improves anticancer efficacy. Thus, in this setting, p300-mediated PD-L1 acetylation may be a favorable factor for immunotherapy. This study was conducted using human BC cell lines in vitro and the mouse MC38 syngeneic colon cancer model [286]. Collectively, it appears that p300 plays multiple roles in regulating PD-L1, including transcriptional activation and post-translational acetylation. Further studies are clearly required to assess therapeutic potential of CBP/p300 inhibitors alone and in combination with immunotherapy in PCa, BC, and other cancer types.

## Figures and Tables

**Figure 1 cancers-13-02872-f001:**
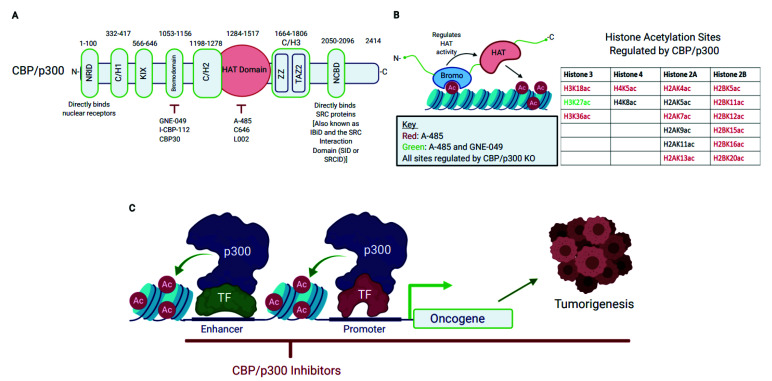
Structure and Function of acetyltransferases CBP/p300. (**A**) Structural modules of CBP/p300. Amino acid numbers are based on the p300 protein sequence. (**B**) Sites of histone acetylation catalyzed by CBP/p300 on the four core histones. All sites are regulated by CBP/p300; sites in red are downregulated by A-485 and sites in green are downregulated by both A-485 and GNE-049. (**C**) CBP/p300 function as co-activators of specific transcription factor. CBP/p300 inhibitors can reduce oncogene expression. Abbreviations: ac: acetylated lysine; Bromo: bromodomain; C/H1, C/H2 and C/H3: cysteine–histidine-rich regions 1, 2 and 3; NRID: the nuclear receptor interaction; HAT: histone acetyltransferase; IBiD: interferon-binding domain; KIX: a kinase-inducible CREB interaction region; NCBD: the nuclear co-activator-binding domain; SID/SRCID: steroid receptor co-activator protein-interaction domain; SRC: steroid receptor co-activator protein; TAZ: transcriptional adaptor zinc-binding domain; TF: transcription factor; ZZ: ZZ-type zinc finger.

**Figure 2 cancers-13-02872-f002:**
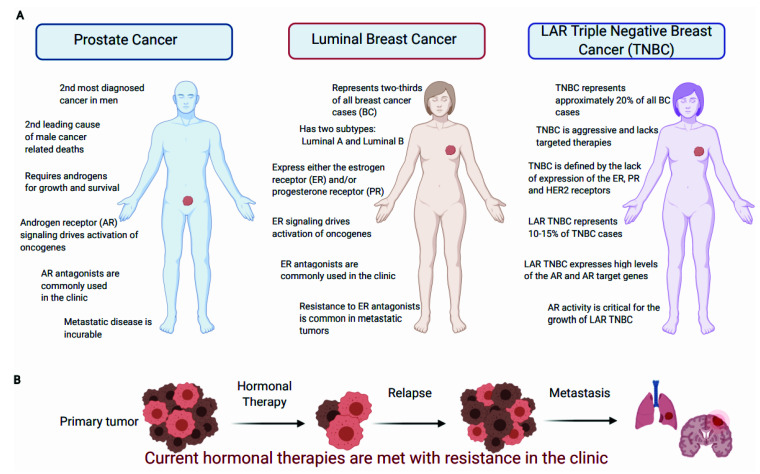
Challenges in Hormone-Dependent Prostate and Breast Cancers. (**A**) An overview of the clinical challenges for hormone-dependent prostate cancer, luminal breast cancer, and LAR (luminal androgen receptor) triple-negative breast cancer. (**B**) Resistance to hormonal therapies is a critical clinical challenge after patients relapse on therapy, as depicted in the diagram.

**Figure 3 cancers-13-02872-f003:**
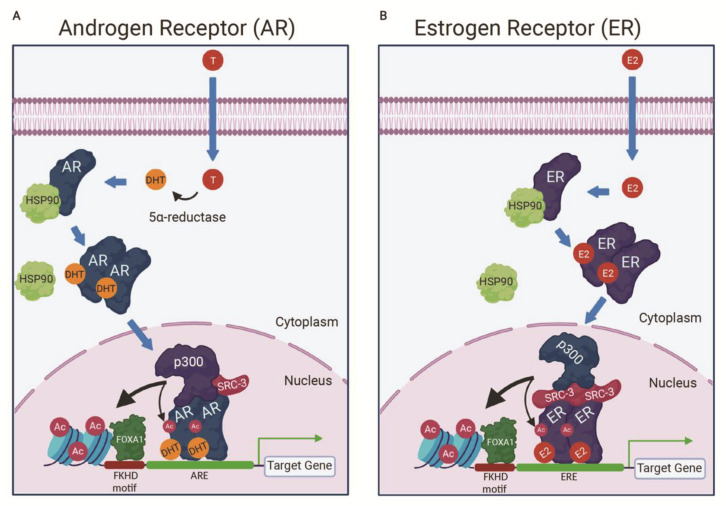
Nuclear Hormone Receptor Signaling. (**A**) An overview of the androgen receptor signaling cascade and formation of the AR activation complex at the ARE in AR target genes. (**B**) An overview of the estrogen receptor signaling cascade and formation of the ER activation complex at the ERE in ER target genes. Ac: lysine acetylation; ARE: androgen response element; DHT: 5α-dihydrotestosterone; E2: 17β-Estradiol; ERE: estrogen response element; FKHD: forkhead motif; HSP: heat shock protein; T: testosterone.

**Figure 4 cancers-13-02872-f004:**
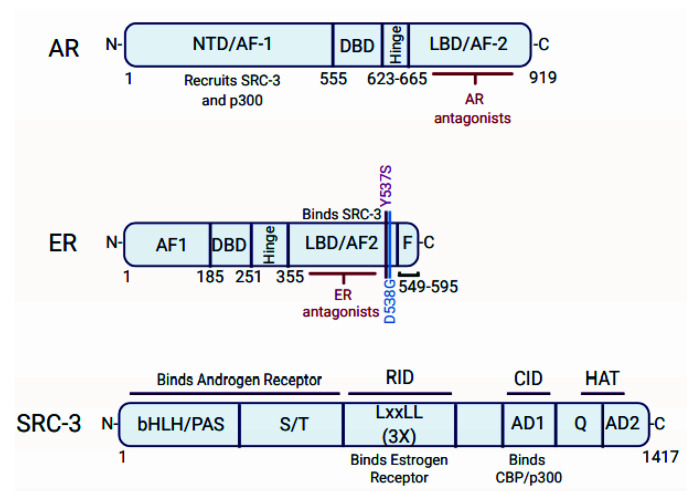
Structure of Androgen and Estrogen Receptors and their Co-activator SRC-3. The structural domains of AR, ER and SRC-3 are shown. The sites of their interactions are depicted. The ligand-binding domain of the AR and ER where antagonists bind is indicated. AD: activation domain; AF: activation function; bHLH/PAS: basic helix-loop-helix/per-arnt-sim domain; CID: CBP/p300 interaction domain; DBD: DNA-binding domain; F: F domain in the C-terminus of nuclear receptors; HAT: histone acetyltransferase domain; LBD: ligand-binding domain; NTD: N-terminal domain; RID: receptor interaction domain; Q: polyglutamate region; ST: serine/threonine rich region.

**Table 1 cancers-13-02872-t001:** Characteristics of the interactions of CREB-binding protein (CBP)/p300 with the androgen receptor (AR) and estrogen receptor (ER).

Biochemical/Functional Properties	AR	ER
CBP/p300 binding domain and interaction features	Direct interaction with p300 via AR NTD and AR LBD [50]	Indirectly interacts with p300 via SRC-3. p300 directly binds SRC-3 in a complex with ER and DNA [51]
Chromatin co-localization with CBP/p300	83% of androgen-induced genes with direct AR binding have overlapping p300 binding [52]	56% (p300) [51] and 38% (CBP) [53] of binding sites are shared with ER. 69% of ER binding sites are co-occupied by CBP and p300 [53]
Protein stability	CBP/p300 acetylates AR and enhances AR stability [11]	CBP/p300 acetylates ER and enhances ER stability [54]
Tumorigenesis	CBP/p300 inhibitors downregulate AR target gene expression and inhibit PCa growth [28,52]	CBP/p300 inhibitors suppress expression of estrogen-regulated genes and block ER+ BC proliferation in vitro [55]

**Table 2 cancers-13-02872-t002:** CBP/p300 inhibitors *.

HAT Inhibitors	In Vitro Potency (IC_50_, nM)	Cellular Activity	In Vivo Effects
A-485	9.8 (p300); 2.6 (CBP) [28]	Reduces H3K27ac and H3K18ac [22,28,55]; suppresses enhancer H3K27ac; represses AR/ER target genes [28,55,85]	Blocks CRPC/MM xenograft growth [28,85]; enhances antitumor efficacy of anti-PD-L1 antibody [86]
iP300w	n/a	Reduces H3K27ac and H3K18ac [87]	Alters DUX4 target gene expression [87]
B026	1.8 (p300); 9.5 (CBP) [88]	Reduces H3K27ac; inhibits MYC expression; inhibits growth AR+ PCa cell lines [88]	Inhibits tumor growth of MV-4-11 AML xenografts [88]
12	620 (p300) and 1200 (CBP) [89]	Reduces H3K27ac, H3K18ac and H3K9ac; suppresses growth of MCF-7 and other cancer cell lines [89]	n/a
21	11 (p300) [90]	Reduces H3K27ac with an EC_50_ of 4 nM; inhibits LNCaP proliferation with an EC_50_ of 17 nM [90]	n/a
**BD inhibitors**			
GNE-049	2.3 (p300); 1.1 (CBP) [52]	Reduces H3K27ac at enhancers [23] and represses AR/ER target genes [52,55]	Blocks CRPC PDX growth and AR target gene expression [52]
GNE-781	1.2 (p300); 0.9 (CBP) [91]	Reduces MYC expression [91]	Blocks AML xenograft growth [91]
CCS1477	19 (p300) [92]	Represses AR target genes [92]	Blocks CRPC xenograft and PDX growth; in clinical trial [92]
**CBP/p300 PROTAC**			
dCBP-1	n/a	Causes proteasomal degradation of CBP and p300; reduces enhancer H3K27ac and chromatin accessibility; downregulates MYC; suppresses in vitro proliferation of MM cells [93]	n/a

* Note: only select small molecule inhibitors reported since 2017 are shown here. n/a: data not available. AML: acute myeloid leukemia; AR: androgen receptor; BD: bromodomain; CBP: CREB-binding protein; CRPC: castration-resistant prostate cancer; ER: estrogen receptor; HAT: histone acetyltransferase domain; MM: multiple myeloma; PCa: prostate cancer; PDX: patient-derived xenograft; PROTAC: proteolysis targeting chimera.

## Data Availability

Not applicable.

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
