# Peer review of "CBP/p300: Critical Co-Activators for Nuclear Steroid Hormone Receptors and Emerging Therapeutic Targets in Prostate and Breast Cancers"

_cancers, 2021, doi:10.3390/cancers13122872_

Round 1
Reviewer 1 Report
Overall the manuscript is quite thorough in summarizing the literature about p300/CBP in context of prostate and breast cancer and its potential as emerging therapeutics. Since this manuscript discusses nuclear steroid receptors in prostate and breast cancer, it would be helpful to discuss the similarities and difference of the roles and therapeutic targeting of p300/CBP. Also, a lot of the text information can be summarized into a table or two.
Minor edits/typos:
Line 910: KAT not kAT
Line 940: PD-L1 nuclear localization not PD-L1 unclear localization
Lines 930-948 could be part of section 4 and 5. This paragraph does quite fit in as a concluding statement
Sections 3.3 and 3.4 can we switched to maintain the flow of discussing AR activation and AR-activated genes followed by discussing p300 in the absence of AR
Reviewer 2 Report
Authors did very nice job.
The review is comprehensive and very well written.
In my opinion this review will be nice contribution for the fields.
Reviewer 3 Report
Dear Authors,
Waddell et al is an excellent and extremely thorough review concerning CBP/p300 as an emerging druggable target for steroid hormone-driven cancers. The major obstacle on the path to recovery for patients affected by prostate and breast cancers is the emergence of treatment-driven resistance mechanism. In this article the authors aim to provide insights into the role of cofactors in cancer development and progression, and exploit AR and ER dependency on CBP/p300. There is a great need in the cancer field to focus not only on a particular type of cancer, but rather to understand the mechanisms that drive the disease and identify predictive biomarkers. This had led to the development of precision oncology where clinicians are focused on genetic features of the cancer rather than the place of origin. The authors clearly understand this, hence, decided to discuss prostate and breast cancers together. This is further justified by identification of triple negative breast cancers with the dependency on AR that can be treated with androgen-deprivation therapies, and conversely, BRCA1/2-deficient prostate cancers that can be treated with PARP inhibitors.
In this article, the authors present a compelling story pointing to CBP/p300 as the main culprit behind the nuclear hormone-driven aberrant gene expression promoting cancer. They provide an overview of the structure and function of these proteins, but also point out the 2 major weaknesses of CBP/p300; their bromo- and lysine acetyltransferase domains. Many potent inhibitors have been developed to inhibit the catalytic CBP/p300 property, which are now being tested in either a pre-clinical or clinical setting. This review offers a well-balanced summary of where the field is regarding PCa and BC treatments and provides a starting point for future discoveries that target nuclear receptor cofactors, thus circumventing a host of resistance mechanisms to current therapies.
Waddell et al is a comprehensive and well-written review, therefore challenging to provide suggestions or identify areas for improvement. One detail worth mentioning that the authors/audience might be interested in, is that CARM1 (described as a component of ER signaling and p300 interaction partner but omitted in the prostate cancer section) has also been found to be required for DHT-dependent AR activation on par with CBP/p300 (Baek et al, 2006 PNAS). Yet another argument for considering these two cancer types together when looking at potential druggable targets.
